# Mechanisms of CYP450 Inhibition: Understanding Drug-Drug Interactions Due to Mechanism-Based Inhibition in Clinical Practice

**DOI:** 10.3390/pharmaceutics12090846

**Published:** 2020-09-04

**Authors:** Malavika Deodhar, Sweilem B Al Rihani, Meghan J. Arwood, Lucy Darakjian, Pamela Dow, Jacques Turgeon, Veronique Michaud

**Affiliations:** 1Tabula Rasa HealthCare Precision Pharmacotherapy Research and Development Institute, Orlando, FL 32827, USA; mdeodhar@trhc.com (M.D.); srihani@trhc.com (S.B.A.R.); marwood@trhc.com (M.J.A.); ldarakjian@trhc.com (L.D.); pdow@trhc.com (P.D.); jturgeon@trhc.com (J.T.); 2Faculty of Pharmacy, Université de Montréal, Montreal, QC H3C 3J7, Canada

**Keywords:** drug–drug interactions, mechanism-based inhibition, competitive inhibition, non-competitive inhibition, substrate, inhibitor, cytochromes P450

## Abstract

In an ageing society, polypharmacy has become a major public health and economic issue. Overuse of medications, especially in patients with chronic diseases, carries major health risks. One common consequence of polypharmacy is the increased emergence of adverse drug events, mainly from drug–drug interactions. The majority of currently available drugs are metabolized by CYP450 enzymes. Interactions due to shared CYP450-mediated metabolic pathways for two or more drugs are frequent, especially through reversible or irreversible CYP450 inhibition. The magnitude of these interactions depends on several factors, including varying affinity and concentration of substrates, time delay between the administration of the drugs, and mechanisms of CYP450 inhibition. Various types of CYP450 inhibition (competitive, non-competitive, mechanism-based) have been observed clinically, and interactions of these types require a distinct clinical management strategy. This review focuses on mechanism-based inhibition, which occurs when a substrate forms a reactive intermediate, creating a stable enzyme–intermediate complex that irreversibly reduces enzyme activity. This type of inhibition can cause interactions with drugs such as omeprazole, paroxetine, macrolide antibiotics, or mirabegron. A good understanding of mechanism-based inhibition and proper clinical management is needed by clinicians when such drugs are prescribed. It is important to recognize mechanism-based inhibition since it cannot be prevented by separating the time of administration of the interacting drugs. Here, we provide a comprehensive overview of the different types of mechanism-based inhibition, along with illustrative examples of how mechanism-based inhibition might affect prescribing and clinical behaviors.

## 1. Introduction

In clinical practice, the need for the use of multiple drugs is common, as patients often present with numerous chronic diseases. To improve commodity and drug adherence, several medications are often administered concomitantly. Although this may represent a preferred clinical strategy, the administration of two or more drugs at overlapping times increases the likelihood of drug–drug interactions [1,2]. As the risk of drug–drug interactions increases, the risk of debilitating, even fatal, adverse drug events also increases [3]. From a pharmacokinetics standpoint, drug–drug interactions occur when one drug—the perpetrator drug—alters the disposition of another co-administered drug—the victim drug.

Inhibition of cytochrome P450 (CYP450) enzymes is the most common mechanism leading to drug–drug interactions [4]. CYP450 inhibition can be categorized as reversible (including competitive and non-competitive inhibition) or irreversible (or quasi-irreversible), such as mechanism-based inhibition. Each type of interaction involves a distinct clinical management strategy. This is why a comprehensive understanding of mechanisms of CYP450-mediated metabolism inhibition is needed to prevent or mitigate these harmful drug interactions. This review will focus on the CYP450 enzymatic system with a special look at one specific type of CYP450 inhibition, namely mechanism-based inhibition; clinical cases involving mechanism-based inhibitors will be discussed in this context.

## 2. Basic Concepts of Enzyme, Substrate, and Inhibitor

For clarity, as we address the various types of drug metabolism inhibition, the concepts of active (orthosteric) and allosteric sites, substrates, and inhibitors need to first be reviewed.

### 2.1. Active Site or Orthosteric Site of the Enzyme

The active or orthosteric site is a physical space or pocket within the protein structure of an enzyme where a molecule can bind and from where a catalytic reaction takes place to convert the molecule into a metabolite (addition of a hydroxyl moiety, removal of alkyl moieties, etc.). (Figure 1) For CYP450 isoforms, binding to the active site is independent of the NADPH-P450 oxidase reactions; however, the chemical reaction leading to the formation of the metabolite will employ electrons originated from NADPH.

### 2.2. Allosteric Site

The allosteric site is a physical space or pocket within the protein structure of an enzyme spatially separated from the active site. (Figure 1) The allosteric site allows molecules to modulate enzyme activity. These molecules can be allosteric activators or allosteric inhibitors, depending on how they influence enzyme activity. Drugs can bind to this site and change the three-dimensional structure of the enzyme (conformational change). Allosteric inhibitors may render the active site no longer accessible for substrate binding or make the site unable to catalyze reactions. It is widely known that almost all cases of non-competitive inhibition are caused by allosteric regulation (see discussion in Section 3).

### 2.3. Substrates

Substrates are drugs that bind to the active site of an enzyme and are transformed into metabolites while being present in this active site. The biotransformation process of a drug may involve multiple enzymes leading to various metabolites; each metabolic route relies on specific characteristics. The strength of attraction between an enzyme and a substrate is measured as the “binding affinity”. A substrate can exhibit varying binding affinity for an active site depending on their chemical structure and physical properties. Based on their binding affinity for a specific enzyme, substrates can be classified into weak, intermediate, and strong affinity substrates. Advanced clinical decision support systems (such as MedWise™) depict the various degrees of affinity by different colors: light yellow (weak), dark yellow (intermediate), and orange (strong affinity).

### 2.4. Binding Affinity

Binding affinity to an enzyme is measured by the *K_m_*, i.e., the concentration at which 50% of the maximum metabolic reaction (*V_max_*) occurs; the lower the *K_m_*, the greater the affinity. The intrinsic clearance measures the ability of an organ to clear unbound drug when there are no limitations to blood flow and binding considerations. The intrinsic clearance (*CL_int_)* of a substrate is defined by:$$CL_{int}=\frac{Vmax}{\left(Km+\left[S\right]\right)}\;$$
where [*S*] is the substrate concentration. In most clinical situations, liver enzymes are rarely saturated so that, generally, the substrate concentration is much smaller than the *K_m_* and the equation can be simplified to:$$CL\;int\;\approx \frac{Vmax}{Km}$$

The binding affinity of a substrate can be modified by the presence of other molecules (in many drug–drug interaction situations, the *K_m_* is increased for the victim drug such that its *CL_int_* is decreased).

### 2.5. Inhibitors

Drugs defined as inhibitors bind either to the active site or to an allosteric site of the enzyme. However, they can also bind to both; in these cases, the process is called “mixed inhibition” and can often be more potent than simple competitive or non-competitive inhibition. Inhibitors can be either substrates or non-substrates of the enzyme. As mentioned previously, non-substrate inhibitors typically bind to an allosteric site of the enzyme. If the inhibitor is a substrate transformed by the enzyme, the substrate itself or its metabolites could contribute to the inhibition mechanism. For example, studies on the inhibitory potency of gemfibrozil indicated that gemfibrozil is a potent inhibitor of CYP2C9 in vitro, but that it is a more potent inhibitor of CYP2C8 than CYP2C9 in vivo [5,6,7]. This observation is substantiated by several clinical reports of interactions between gemfibrozil and CYP2C8 substrates including cerivastatin, repaglinide, and glitazones [8,9,10,11]. The mechanism of this clinical interaction is explained by the formation of the major metabolite of gemfibrozil, gemfibrozil 1-O-β-glucuronide, which was found to potently inhibit CYP2C8 [10,12].

## 3. Mechanism of CYP450 Inhibition

Drug interactions due to drug metabolism inhibition are frequent since (1) CYP450-mediated metabolism is the major route of elimination for a large number of drugs, and (2) multiple drugs can compete for the same CYP450 active site. Mechanisms of CYP450 inhibition can be categorized as reversible (including competitive or non-competitive) or irreversible/quasi-irreversible (mechanism-based inhibition).

### 3.1. Reversible CYP450 Inhibition

Reversible inhibition is a result of rapid association and dissociation between the substrate drugs and the enzyme and can be categorized as competitive or non-competitive. A third category, uncompetitive inhibitor, also considered as a reversible inhibition type, is a very rare phenomenon and will not be considered in this current review; this type of inhibitor binds only the enzyme–substrate complex, leading to a dead-end complex.

#### 3.1.1. Competitive Inhibition

The ability of a single CYP450 isoform to metabolize multiple substrates is responsible for several drug interactions associated with reversible competitive inhibition. Competitive inhibition occurs when two substrates compete for the same active site—such as the prosthetic heme iron or substrate-binding region—of CYP450s. The competition is a function of the respective affinities of the two substrates for the binding site and their concentrations in the proximity of the enzyme. First, the most clinically relevant situation will be discussed.

##### Two Substrates with Different Affinities Administered Concomitantly

This situation is often encountered in clinical practice. Under a competitive inhibition condition, a substrate with strong affinity (acting as a perpetrator) can displace a weaker substrate (behaving as a victim) from the active site (Figure 2), thus increasing the *K_m_* of the victim drug (decreased affinity) and reducing the extent of its breakdown (decrease in its *CL_int_*) over a period of time.

For an active drug, the decrease in the *CL_int_* of one of its metabolic pathways can lead to a decrease in the total clearance of the drug (capacity for eliminating the drug) and can result in increased plasma concentrations, potentially precipitating adverse effects. However, for prodrugs, this interaction can instead result in a decrease in the formation of the active metabolite, reducing drug efficacy. The magnitude of changes observed in the overall disposition of the victim drug (increase in its plasma levels) will be a function of the relative contribution of the inhibited metabolic pathway to the clearance of this drug. For example, if 15% of a drug is excreted unchanged in urine—35% by enzyme 1 and 50% by enzyme 2—a 50% decrease in the total *CL* of the victim drug is expected if enzyme 2 is inhibited:*CL* = *CL_renal_* + *CL_metabolic_* = 0.15 + 0.85
and,
*CL_metabolic_* = *CL*_*enzyme* 1_ + *CL*_*enzyme* 2_
or,
0.85 = 0.35 + 0.5
under conditions of enzyme 2 inhibition (whether it is reversible or irreversible),
*CL_metabolic_* = *CL*_*enzyme* 1_ + CL_*enzyme* 2_
or,
0.35 = 0.35 + 0.0
and,
*CL* = *CL_renal_* + *CL_metabolic_* = 0.15 + 0.35 = 0.5

Since,
*CL* = *Dose*/*AUC*_0–∞_

Under steady-state conditions, the area under the drug concentration curve (*AUC*) measured over a dosing interval (τ) is equal to *AUC*_0–∞_. Since the average concentration over a dosing interval (*C_av_*) at steady state can be estimated by *AUC*_0-*τ*_/*τ*, the equation could be rearranged in a simpler manner to yield:*CL* = *Dose*/(*C_av_* × *τ*) 

A 50% decrease in *CL* will be associated with a doubling in the average plasma concentrations of the victim drug.

According to a competitive inhibition mechanism, every substrate of an enzyme is a potential perpetrator drug (inhibitor) towards another substrate metabolized by the same enzyme. Competitive inhibition is almost immediate and the degree of inhibition of the enzyme does not change with time if the concentration of the two substrates remains the same.

In an example illustrating this scenario, theophylline (weak CYP1A2 substrate) is largely metabolized by CYP1A2 by binding to its active site. (Figure 3) If that active site is occupied by a stronger substrate like duloxetine (moderate CYP1A2 affinity substrate), breakdown of theophylline will be reduced (↓*CL_int_*), leading to increased plasma levels of theophylline and possibly side effects (e.g., headache, nausea, vomiting). To minimize competitive inhibition, two competing substrates should be administered with as much time apart as possible.

##### Two Substrates with Largely Different Concentrations

As mentioned previously, the competitive inhibition process is sensitive to substrate concentrations. If concentrations of the weaker affinity substrate are much higher than concentrations of the stronger affinity substrate, the weaker affinity substrate can displace the stronger affinity substrate and overcome the enzyme inhibition, which is why this type of inhibition is deemed reversible. (Figure 4) The greater the difference there is between the affinity of the weaker affinity substrate and the stronger affinity substrate, the more the concentration of the weaker affinity substrate needs to be increased to displace the stronger affinity substrate. This situation can be observed clinically when very high concentrations of a weak affinity substrate are present in the intestine or liver (high micromolar concentrations), following its oral administration, while concentrations of another higher affinity substrate have long been absorbed and distributed to various tissues leading to plasma concentrations in the low nanomolar range. In this case, the extent of victim drug inhibition would be minimal. A direct application of this principle is to alleviate the degree of inhibition by separating the time of administration of the two competing drugs.

#### 3.1.2. Non-Competitive Inhibition

The non-competitive inhibitor does not generally have any structural resemblance to the substrate as it binds to an allosteric site. The non-competitive inhibitor will cause a conformational change in the structure of the active site such that the active site loses its affinity for the substrate. (Figure 5) Thus, there is no direct competition between the inhibitor and the substrate at the active site. This type of inhibition is often long lasting and cannot be overcome by increasing substrate concentrations. Under these conditions, a decrease in the *CL_int_* of the substrate due to a decrease in its *V_max_* is observed. Similar to competitive inhibitors, non-competitive inhibitors also have an almost immediate effect. As long as the concentration of the inhibitor is not changed, the amount of inhibition will not increase over time. This type of inhibition does not require the involvement of NADPH as a cofactor, i.e., the inhibitor is not metabolized by the enzyme, but merely sits in an allosteric site. Other non-competitive inhibition conditions may involve CYPb5 and/or CYP450 oxidoreductase as these factors have been shown to modulate CYP450 activities, at least in in vitro systems [13,14]. Separating the time of dosing will not alleviate non-competitive inhibition. Fluvoxamine (CYP2C19) and terbinafine (CYP2D6) are some common examples of non-competitive inhibitors at other CYP isoforms [15,16,17].

#### 3.1.3. Mixed inhibition

In the case of mixed inhibition, both competitive and non-competitive inhibition occur. Mixed inhibitors can simultaneously bind to both the heme iron atom (at the active site) and lipophilic regions of the protein (allosteric site). Mixed inhibitors are usually more potent inhibitors than competitive or non-competitive inhibitors. Ketoconazole and fluconazole, both imidazole antifungals, exhibit potent mixed reversible inhibition of CYP3As. However, fluconazole is a weaker mixed reversible inhibitor compared to ketoconazole, mainly due to its lower lipophilicity (less binding to an allosteric site).

For a CYP3A substrate like midazolam, concomitant use of non-competitive or mixed CYP3A inhibitors will reduce its transformation to α-hydroxy midazolam, increasing midazolam plasma levels and augmenting the risk of adverse drug events [18].

### 3.2. Irreversible CYP450 Inhibition

Several clinically important pharmacokinetic drug interactions result from a decrease in the metabolic clearance of a substrate due to CYP450 irreversible inhibition. Mechanism-based inhibition is a condition often encountered with irreversible CYP450 inhibitors.

#### 3.2.1. Mechanism-Based Inhibition

Mechanism-based inhibition can be irreversible or quasi-irreversible. It generally derives from the activation of a substrate drug by a CYP450 isoform into a reactive metabolite, which binds to the enzyme heme prosthetic site (part of the active site), resulting in irreversible long-lasting loss of enzyme activity (decrease in *V_max_)*. (Figure 6) Several drugs undergo metabolic activation by a specific CYP450 isoform to produce inhibitory intermediate metabolites, which can form stable intermediate complexes. As a result, the CYP450 isoform is sequestered in an inactive state. Even though the reactive intermediate metabolite plays a key role in the mechanism-based inactivation of the CYP450 isoform, in many instances, the exact reactive metabolite involved in this phenomenon is unknown.

In the case of quasi-irreversible inhibition, the metabolites form very stable complexes with the heme prosthetic site (metabolite–intermediate complex), so that the enzyme is sequestered in a functionally inactive state. This phenomenon is called quasi-irreversible since, in theory, this complex can be disrupted. In the case of irreversible inhibition, the metabolites covalently bind to the heme prosthetic site or the protein part of the CYP450, leading to irreversible inactivation [19,20].

Hence, mechanism-based inhibition is active site mediated, and the allosteric site is not involved. In contrast to reversible inhibition mechanisms, mechanism-based inhibition is time dependent and NADPH dependent. This means that the enzyme has to start breaking down the substrate in order for inhibition to proceed. As more drug molecules are metabolized, more complexes are stably formed in the active sites, increasing inhibition over time before it reaches a plateau. Mechanism-based inhibition is therefore also saturable. New enzyme formation is necessary to restore activity: the relationship between the amount of intermediate complex formed and the speed of new enzyme synthesis dictate the equilibrium and extent of enzyme inhibition.

Mechanism-based inhibitors can be classified into two categories: metabolic–intermediate complex formation inhibitors and protein and/or heme alkylation inhibitors.

##### Metabolic–Intermediate Complex Formation (or Alternate Substrate Inhibition)

Such a condition occurs when a stable intermediate metabolite formed during the normal metabolic cycle forms covalent bonds at the active site. This stable intermediate–enzyme complex is not easily broken by increasing substrate concentration. Since the enzyme structure remains otherwise unchanged, theoretically this reaction is reversible with time. However, in in vivo conditions, with this metabolic intermediate complex being excessively stable, the metabolic intermediate cannot be displaced and the enzyme remains inaccessible for metabolism so the reaction seems irreversible.

An example of alternate substrate inhibition is observed with paroxetine as its methoxy diene carbon moiety was found to be responsible for the formation of covalent bonds at the active site of CYP2D6 [21,22]. Another example of this type of inhibition was observed with clarithromycin when the nitrosoalkene intermediate generated by N-demethylation forms covalent bonds with the active site of CYP3A4 [23].

##### Protein and/or Heme Alkylation (or Suicide Inhibition)

This situation takes place when a latent highly reactive (generally electrophilic) intermediate is formed in the catalysis process. The reactive intermediate forms covalent bonds (strong irreversible bonds) with the enzyme in a step that is not part of the normal metabolic pathway. This process can change the conformational structure of the enzyme significantly—it can even destroy the enzyme in some cases—making it functionally unviable. For example, inhibition of CYP2C19 by esomeprazole was found to be mediated by crosslinking the heme and apoprotein moieties in the enzyme, changing its conformational structure [24].

It is important to note that since mechanism-based inhibitors are substrates of the enzyme, they can also cause acute competitive inhibition when co-administered with other sensitive substrates. The difference between competitive inhibition and mechanism-based inhibition is that as the time period of exposure to mechanism-based inhibitors increases, the degree of inhibition also increases. (Table 1)

## 4. Clinical Cases

### 4.1. The Case of Omeprazole and Clopidogrel

Proton pump inhibitors (PPIs) are commonly prescribed along with antiplatelet drugs like clopidogrel to reduce the incidence of gastric bleeding during treatment with antiplatelet therapy [25,26]. Omeprazole has long been one of the most widely used PPIs [27]. Omeprazole is a known strong affinity substrate of CYP2C19, leading to the formation of its hydroxy and desmethyl metabolites. The antiplatelet drug clopidogrel is sequentially activated by CYP450 isoforms, including CYP2C19, into its active metabolite (H4) [28]. When omeprazole is co-administered with clopidogrel, omeprazole acts as a strong affinity substrate of CYP2C19 (perpetrator), whereas clopidogrel is a weaker sensitive substrate (victim). (Figure 7) Multiple in vitro studies have reported a potential pharmacokinetic interaction between omeprazole and clopidogrel [29,30,31,32]. A clinical study conducted by Angiolillo et al. demonstrated that plasma levels of clopidogrel’s active metabolite H4, and consequently the platelet aggregation induced by adenosine diphosphate, were decreased when omeprazole and clopidogrel were administered concomitantly [30]. Various other clinical studies have demonstrated that co-administration of omeprazole and clopidogrel diminishes the antiplatelet activity of clopidogrel [31,32].

Since omeprazole is a strong affinity substrate for CYP2C19, an “immediate” competitive inhibition is expected between these two drugs. Since competitive inhibition was expected, separating time of administration was considered a logical mitigation strategy to avoid or alleviate the extent of the drug interaction [29]. Others have suggested that increasing the dose of clopidogrel might compensate for the diminished formation of the active metabolite [29]. These recommendations to separate the time of administration of the two drugs or to increase the dose of the victim drug (clopidogrel) come from a sound rationale and have been proven to be efficacious in mitigating drug interactions associated with competitive inhibition. However, it has been shown that following chronic administration, separating the time of administration does not alleviate the reduction in clopidogrel active metabolite (H4) caused by omeprazole [30,33]. This is due to the fact that omeprazole is not only a competitive inhibitor, but also a mechanism-based inhibitor of CYP2C19, which results in a gradual increase in irreversible inhibition of the CYP2C19 enzyme, to a point where clopidogrel activation and its clinical efficacy are significantly impaired. From these observations, the FDA warns that separating the time of administration between these two substrates will not alleviate this interaction [34].

Multiple studies have been conducted to determine the clinical impact of the potential reduced antiplatelet efficacy resulting from this interaction. In two retrospective studies looking at interactions between clopidogrel and PPIs and the effects on clinical outcomes, it was reported either that PPIs were associated with increased cardiac adverse events in acute coronary syndrome patients, or that cardiac adverse events were less common in PPI non-users [35,36]. Short-term mortality odds ratios also favored PPI non-users, but no significant differences were observed in long-term mortality [35]. Though a wide range of PPIs were reviewed, omeprazole and esomeprazole remained the most widely prescribed when all studies were combined. A similar study was conducted by Mahabaleshwarkar et al., which found that PPIs were slightly, but significantly, associated with all-cause mortality [37]. The odds ratio of adverse cardiac events and all-cause mortality for omeprazole in particular was 1.23. In another retrospective cohort study, clopidogrel use post discharge for acute coronary syndrome hospitalizations was studied [38]. Concurrent clopidogrel and PPI use was associated with an increased risk of death or rehospitalization; among patients prescribed a PPI, 60% were on omeprazole [38]. PPI plus clopidogrel use also remained significantly associated with recurrent acute coronary syndrome and revascularization procedures [38]. Another study evaluated the association between various PPIs (all PPIs combined) and individual PPI agents with clopidogrel use and increased risk of hospitalization [39]. There was no significant association between any PPI and increased risk of rehospitalization with clopidogrel, but this association was significant with omeprazole [39].

Several studies also report no change in the frequency of cardiovascular adverse events with omeprazole administration during clopidogrel treatment [26,40,41,42]. Dosing regimens in these studies suggest that the extent of interaction between clopidogrel (75 mg vs. 600 mg) and omeprazole (20 mg vs. 80 mg) may be dose dependent. If an alternative PPI has to be considered, in vitro studies using human liver microsomes have confirmed that PPIs like rabeprazole, lansoprazole, dexlansoprazole, and pantoprazole do not show evidence of mechanism-based inhibition [29]. Clinical studies have also reported that effects of lansoprazole and pantoprazole on clopidogrel antiplatelet activity are not as potent as omeprazole [43,44,45]. When clopidogrel and PPI coadministration is necessary, switching to a PPI other than omeprazole or esomeprazole may be considered, in light of the evidence presented herein.

### 4.2. The Case of Paroxetine

The antidepressant paroxetine is a known substrate of CYP2D6, but also a potent mechanism-based inhibitor of this enzyme. As such, paroxetine is expected to inhibit its own metabolism over time. This has been illustrated in two clinical studies, where the *C_max_* and AUC of paroxetine were increased 5.2- and 7-fold after 2 weeks of paroxetine administration, respectively [46,47]. In the second study, Laine et al. revealed that ultra-rapid metabolizers of CYP2D6 were converted to extensive or poor metabolizers with chronic paroxetine use [47]. The FDA-approved label states that paroxetine takes nearly 10 days to achieve steady-state concentrations even though the drug has a reported elimination half-life of 21 h; so, within 4–5 days, steady-state levels should be reached under normal circumstances. The label also states that saturation of CYP2D6 contributes to the non-linear pharmacokinetics of paroxetine. Thus, it may be assumed that inhibition of its own mechanism contributes to achieving later than expected steady-state levels and clinical efficacy [48].

As paroxetine is a strong CYP2D6 affinity substrate, it can exhibit acute competitive inhibition when co-administered with sensitive substrates like nortriptyline. (Figure 8) Though separating the time of administration may seem appropriate initially, the effects of mechanism-based inhibition over time should be factored into medication risk management; dosage adjustment or substitution of the victim substrate may be necessary. The extent of this interaction also depends on when paroxetine and the victim drug are added to the regimen. If paroxetine is newly added to a victim drug that has already reached steady-state plasma levels, the extent of inhibition and plasma concentrations of the victim drug will increase over time until reaching a new steady state. However, if steady-state levels of paroxetine are already achieved before adding another sensitive substrate, the extent of inhibition will be maximum at initiation and will remain stable, since saturation of enzyme inhibition is already established.

### 4.3. The Case of Erythromycin

Another example that illustrates the concept of saturability in mechanism-based inhibition is observed with the commonly used macrolide antibiotics erythromycin (Figure 9) or clarithromycin. These drugs are known to cause mechanism-based inhibition of CYP3A4. Clinical studies have demonstrated saturability (i.e., degree of inhibition reaches a maximum value) of enzyme inhibition using CYP3A4-sensitive substrates like alfentanil or midazolam with or without erythromycin pretreatment (treatment or control, respectively). In a study with healthy males after single or multiple oral dose(s) of erythromycin 500 mg, the effects on alfentanil pharmacokinetics were measured [49]. A 25% increase of alfentanil half-life was observed following a single erythromycin dose compared to control. After a 7-day pretreatment with erythromycin, the half-life of alfentanil was further increased by 25% (up to 56% compared to control), suggesting an increase in inhibition with time [49]. Similar effects were seen on clearance. Although a direct association of this potential drug interaction on clinical outcomes has not been systematically reported, two case reports suggest that erythromycin pretreatment may cause prolonged respiratory depression when alfentanil is administered compared to patients who did not receive erythromycin [50,51].

In another study with 12 healthy volunteers, the effects of erythromycin on midazolam metabolism were studied [52]. It was observed that the *AUC* of midazolam increased 2.3-fold after 2 days of erythromycin pretreatment, compared to control. Following 4 days of pretreatment, midazolam’s *AUC* increase was 3.38-fold. After a 7-day pretreatment, a similar increase (3.38-fold) was observed, indicating an increase in inhibition with repeated administration of erythromycin and that a plateau effect had been reached after 4 days of exposure [52].

### 4.4. The Case of Mirabegron

A widely prescribed drug in the treatment of overactive bladder, mirabegron, also displays characteristics of mechanism-based inhibition for CYP2D6. (Figure 10) The difference in the degree of metabolism inhibition between a competitive inhibitor and a mechanism-based inhibitor is perceived when they are compared using the same victim drug. One study investigated the victim drug desipramine, with the potential competitive inhibitor duloxetine and the mechanism-based inhibitor mirabegron. In this study, duloxetine (moderate CYP2D6 affinity substrate) 30 mg twice a day was administered for 10 days, after which desipramine (weak CYP2D6 affinity substrate) 50 mg was administered as a single dose. Here, duloxetine would act as a competitive inhibitor and desipramine as a victim drug. Accordingly, a 1.2-fold increase in *AUC* and 0.6-fold increase in *C_max_* of desipramine was observed [53]. In a similar study design, desipramine was administered with or without mirabegron pretreatment. First, desipramine was administered alone and, after a washout period, mirabegron (100 mg) was administered for 13 days. On the fourteenth day, mirabegron 100 mg and desipramine 50 mg were co-administered. A 3.41-fold increase in the desipramine *AUC* was observed with mirabegron pretreatment compared to control [54]. This increase in the *AUC* of desipramine was much larger than the increase observed with duloxetine, a “purely” potential competitive inhibitor. It is important to note that in the short term, mechanism-based inhibitors can act as competitive inhibitors if the other drug has lower affinity for the metabolizing enzyme. Duloxetine and mirabegron are both substrates of CYP2D6 with moderate affinity, exhibiting potential competitive inhibition over desipramine; therefore, following a single dose of each drug, a similar level of CYP2D6 inhibition is expected towards the CYP2D6 victim drug. (Figure 10) The higher inhibition observed with multiple doses of mirabegron versus single dose of duloxetine is explained by the mechanism-based inhibition observed over time. Similar effects of chronic mirabegron administration on metoprolol (a weak CYP2D6 substrate) pharmacokinetics are also reported [54]. A 3.3-fold increase in the metoprolol AUC was observed following a pretreatment of 5 days with mirabegron in CYP2D6 normal (i.e., previously called “extensive”) metabolizer subjects [55]. The coadministration of quinidine (a potent CYP2D6 inhibitor) with metoprolol was associated with a similar magnitude of increase in metoprolol’s AUC after a single dose [56]. The intensity of drug–drug interaction through “purely” competitive inhibition is expected to be lower between substrates compared to quinidine’s inhibition; therefore, the mechanism-based inhibition property of mirabegron can explain why the magnitude of drug–drug interaction observed between mirabegron and metoprolol is similar to that observed between quinidine and metoprolol.

In addition to the case examples discussed above, Figure 11 provides a list of CYP450 mechanism-based inhibitors, along with the enzyme inhibited and relevant CYP450 pathways involved in their metabolism. This list is not exhaustive, but provides a quick reference for commonly used medications.

In addition to drug–drug interactions, high variability in terms of CYP450 expression and/or activities can be explained by genetic polymorphisms in genes encoding specific isoforms (such as *CYP2C9*, *CYP2C19*, and *CYP2D6*). This variability on CYP450 expression/activities translates into intersubject variability in drug disposition and drug response. Often, the impact of genetic polymorphisms and drug–drug interactions on CYP450s have been studied separately. However, an interaction exists between these factors. Genetic polymorphisms could also contribute to variability observed in the magnitude of drug–drug interactions observed between two drugs. So, genetic polymorphisms in drug-metabolizing enzymes can affect the occurrence of phenoconversion induced by drug inhibitors. As reported by Storelli et al., differences in CYP2D6 inhibition observed in vitro with paroxetine (mechanism-based inhibitor) or duloxetine (competitive inhibitor) across *CYP2D6* genotypes were not related to their inhibition parameters but likely due to a differential level of functional enzymes as a function of the *CYP2D6* genotype [57,58].

## 5. Conclusions

Polypharmacy in many cases is deemed to be required and elderly patients are particularly prone to this phenomenon. Aging is associated with the presence of multiple independent chronic diseases and is almost always accompanied by multiple drug regimens. Polypharmacy has been associated with many adverse clinical outcomes, such as drug–drug interactions, leading to adverse drug events. Among these, mechanism-based inhibitor–victim drug combinations like clopidogrel and omeprazole are commonly prescribed. Drugs for overactive bladder like mirabegron and tricyclic antidepressants like paroxetine are also commonly prescribed in elderly populations. Polypharmacy is not necessarily synonymous with inappropriate treatment, but in several situations, it can lead to significant drug–drug interactions especially in the presence of mechanism-based inhibitor drugs, as described in the current review. In these cases, polypharmacy might cause problems like blunted efficacy of clopidogrel due to the co-administration of omeprazole, or increased toxicity of other drugs co-administered with paroxetine or mirabegron. Clinicians must be able to recognize and intervene appropriately based on the mechanism of these interactions.

As highlighted in this current review, mechanism-based inhibition can cause severe clinical interactions. The magnitude of these interactions and their impacts depend on the duration of the mechanism-based inhibitor exposure and the exact time-point when the victim drug is introduced into the drug regimen. Without a comprehensive understanding of this mechanism, healthcare professionals might find it difficult to diagnose and mitigate these interactions. Separating the time of administration of interacting drugs cannot mitigate an interaction caused by mechanism-based inhibition. A careful titration of dose, monitoring of clinical effects, or switching to an alternate drug with a different metabolic pathway may become necessary to avoid such interactions. Advanced clinical decision support systems that consider and distinguish competitive versus mechanism-based inhibition drug–drug interactions would help identify potential interactions mediated by enzyme inhibition. Using such tools can help pharmacists quickly identify and mitigate drug interactions, thus helping reduce preventable drug interactions.

## Figures and Tables

**Figure 1 pharmaceutics-12-00846-f001:**
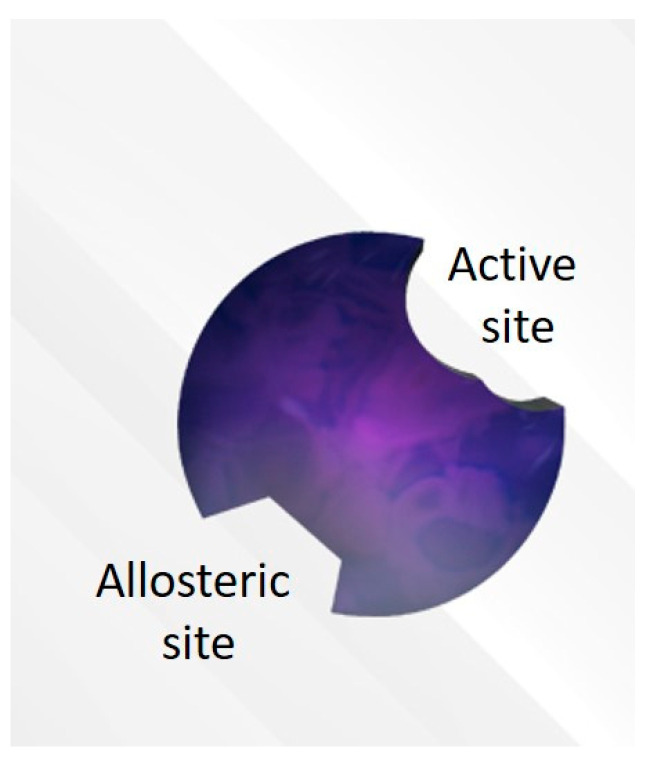
Illustration of an enzyme with its active binding site for drug transformation and allosteric binding site (or regulatory site).

**Figure 2 pharmaceutics-12-00846-f002:**
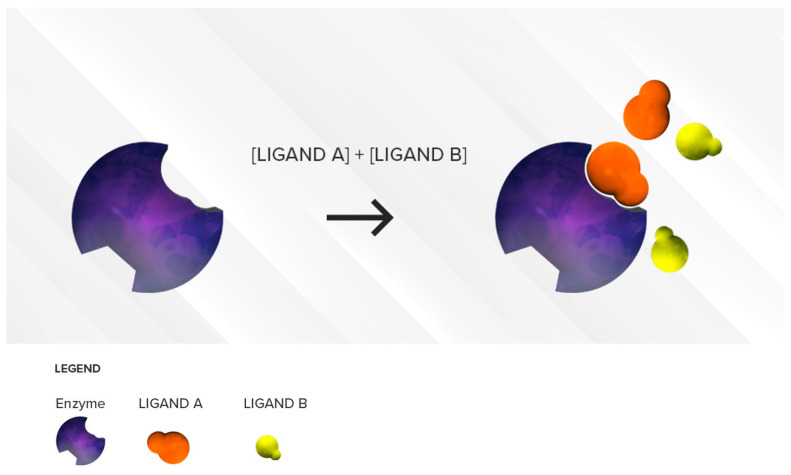
Illustration of reversible competitive inhibition where ligand A (**orange**) is a substrate with strong affinity and ligand B (**yellow**) is a substrate with a weaker affinity for a specific enzyme (**purple**). As long as the concentrations of the two substrates are comparable, the stronger affinity substrate with higher binding affinity will be preferred at the active site of the enzyme resulting in an accumulation of ligand B.

**Figure 3 pharmaceutics-12-00846-f003:**
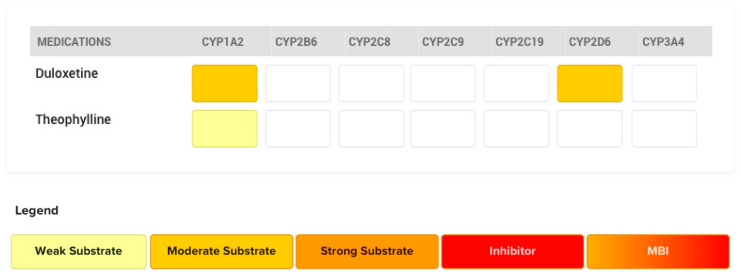
CYP450 metabolic pathways involved in the metabolism of duloxetine and theophylline and their respective affinities are depicted. Competitive inhibition between duloxetine and theophylline will be expected at CYP1A2. Duloxetine acts as the perpetrator drug over theophylline, the victim drug.

**Figure 4 pharmaceutics-12-00846-f004:**
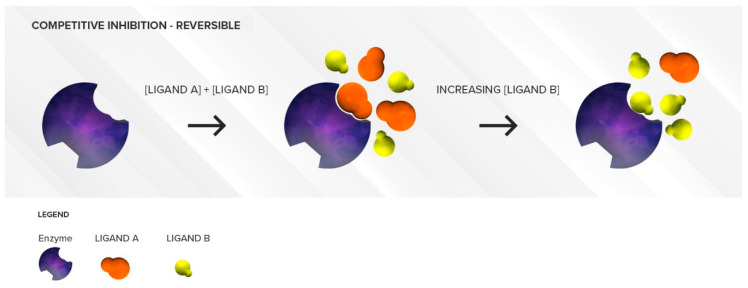
Illustration of reversible competitive inhibition where ligand A (**orange**) is a substrate with strong affinity and ligand B (**yellow**) is a substrate with weaker affinity for a specific enzyme (**purple**). When the concentrations of the weaker affinity substrate are sufficiently high, it can outcompete the stronger affinity substrate for the active site of the enzyme.

**Figure 5 pharmaceutics-12-00846-f005:**
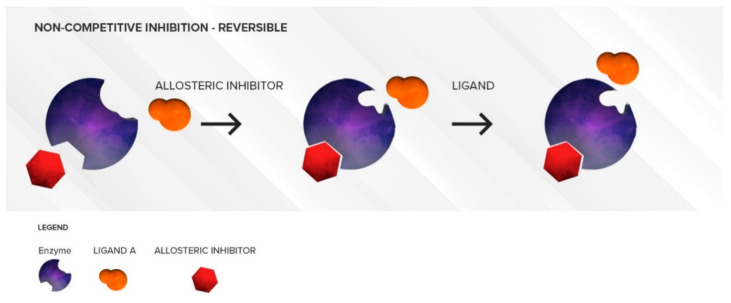
Illustration of reversible non-competitive inhibition. An inhibitor (**red**) binds to an allosteric site on the enzyme and causes conformational changes that prevent a substrate (**orange**) from binding to the active site. Over time, as the inhibitor is flushed out, the conformation of the enzyme can return to normal and substrate (**orange**) can bind to the active site again.

**Figure 6 pharmaceutics-12-00846-f006:**
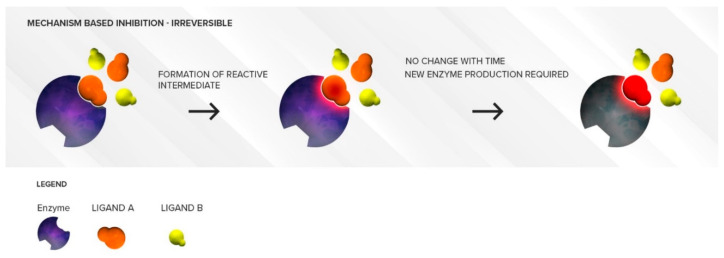
Illustration of mechanism-based inhibition. The mechanism-based inhibitor (**orange**) binds to the active site as a substrate. During the normal process of metabolism, it forms either stable intermediate–enzyme complexes or reactive electrophilic species that can lock up or destroy the enzyme, and new enzyme synthesis is required to restore the enzymatic activity.

**Figure 7 pharmaceutics-12-00846-f007:**
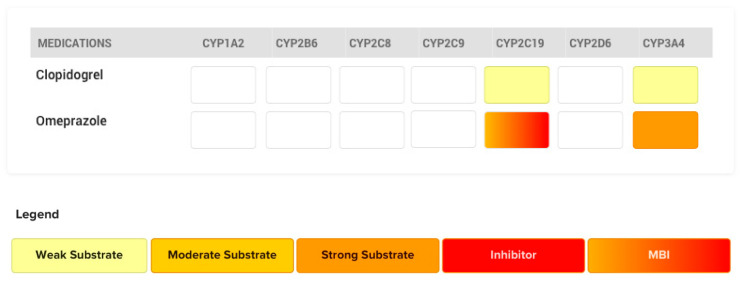
CYP450 metabolic pathways involved in the metabolism of clopidogrel and omeprazole, and their respective affinities are depicted. Competitive inhibition will be expected at CYP3A4 and mechanism-based inhibition at the CYP2C19 enzymatic level. Clopidogrel is the victim drug and omeprazole acts as the perpetrator drug.

**Figure 8 pharmaceutics-12-00846-f008:**
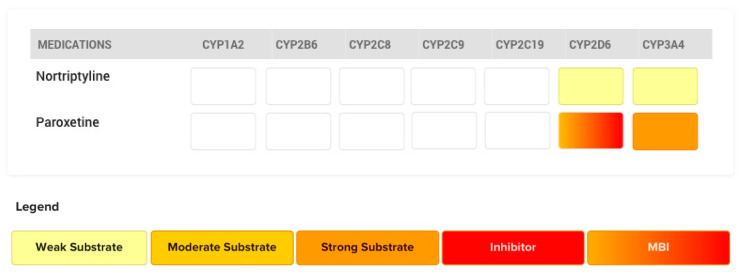
CYP450 metabolic pathways involved in the metabolism of nortriptyline and paroxetine and their respective affinities for the isoform are depicted. Mechanism-based inhibition at CYP2D6 enzymatic level will be expected. Nortriptyline is the victim drug and paroxetine acts as the perpetrator drug for the CYP2D6 elimination pathway.

**Figure 9 pharmaceutics-12-00846-f009:**
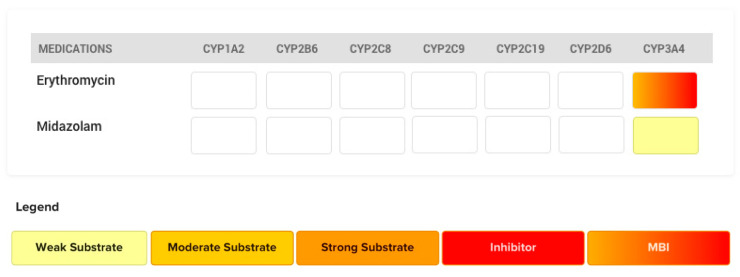
CYP450 metabolic pathways involved in the metabolism of erythromycin and midazolam and their respective affinities for the isoform are depicted. Mechanism-based inhibition at the CYP3A4 enzymatic level will be expected. Midazolam is the victim drug and erythromycin acts as the perpetrator drug.

**Figure 10 pharmaceutics-12-00846-f010:**
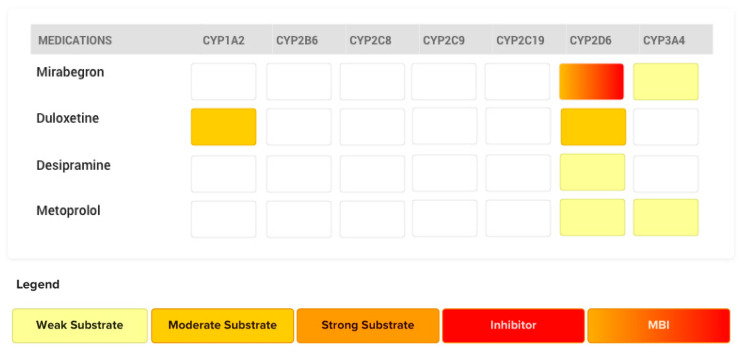
CYP450 metabolic pathways involved in the metabolism of desipramine, duloxetine, metoprolol, mirabegron, and their respective affinities for the isoform are depicted. Competitive inhibition will be expected at CYP2D6 between duloxetine (perpetrator; CYP2D6 substrate with higher affinity) and either desipramine or metoprolol (victim drugs; both CYP2D6 substrates with weaker affinity). Mechanism-based inhibition at CYP2D6 will be expected between mirabegron and desipramine, metoprolol, or duloxetine.

**Figure 11 pharmaceutics-12-00846-f011:**
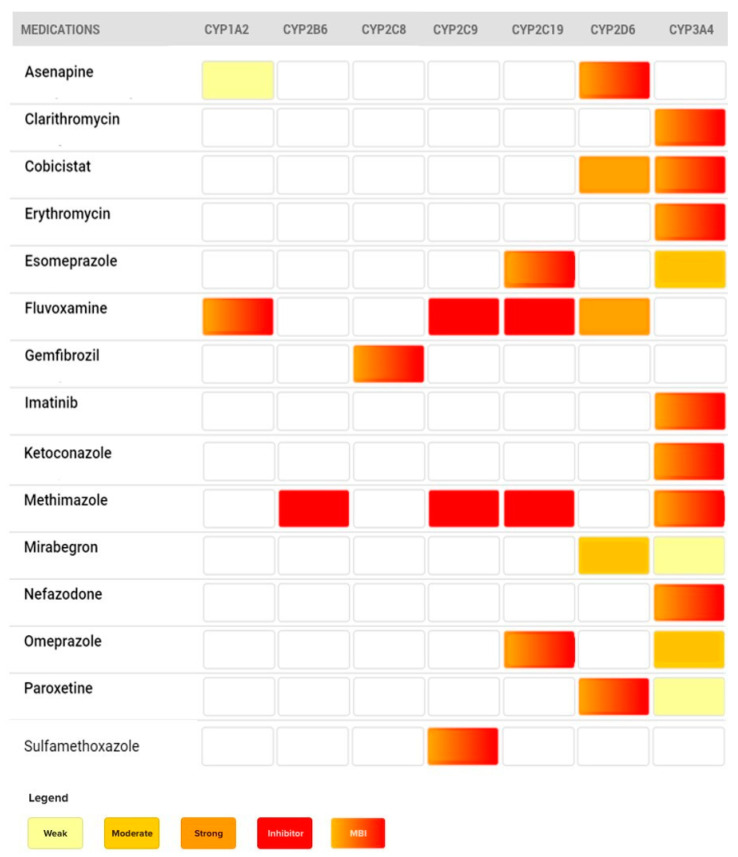
List of commonly prescribed medications producing mechanism-based inhibition.

**Table 1 pharmaceutics-12-00846-t001:** Summary of major pharmacokinetic characteristics of various drug inhibition models.

Characteristics	Inhibitor Type		
	Competitive	Non-Competitive Non-Mechanism Based	Mechanism-Based
Metabolism required	No	No	Yes
Active site mediated	Yes	No	Yes
Time dependent	No	No	Yes
Substrate concentration dependent	Yes	No	Yes
*K_m_* (victim drug)	↑	↔	↔
*V_max_* (victim drug)	↔	↓	↓
*CL_int_* (victim drug)	↓	↓	↓

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
