# Peer review of "Mechanisms of CYP450 Inhibition: Understanding Drug-Drug Interactions Due to Mechanism-Based Inhibition in Clinical Practice"

_pharmaceutics, 2020, doi:10.3390/pharmaceutics12090846_

Round 1

Reviewer 1 Report

The manuscript provided fundamentals and examples on P450 inhibition. It will be good educational material for students and practitioners.

  • The graphs make things easy to understand. For Figures 3, 7, 8… with the panel of enzymes, it will be good to label which one is substrate and which one is inhibitor.
  • For the example in Pages 4-5, it will be good to mention it is an IV drug, since gut DDI is not considered.
  • Equation (9), it should be CL = Dose / AUC

Author Response

We thank the reviewer for taking the time to review and provide positive and productive comments. We have addressed these comments and made appropriate changes:

  • The manuscript provided fundamentals and examples on P450 inhibition. It will be good educational material for students and practitioners.

We thank the reviewer for this comment. Although other publications exist on this topic, they are often presented at a high level for experts in the area and lack a needed didactic nature. In preparing this manuscript, we have taken good care to present scientifically valid information and support the concepts with examples.

  • The graphs make things easy to understand. For Figures 3, 7, 8… with the panel of enzymes, it will be good to label which one is substrate and which one is inhibitor.

We understand the point raised by the reviewer. For clarity, the legend has been modified and enlarged. Since these figures present all metabolic pathways for each drug, a specific “label” indicating whether a drug is a substrate or inhibitor would not present accurate information. In other words, a drug can be the substrate of a CYP450 isoform but a mechanism-based inhibitor towards another CYP450 isoform. For example, omeprazole (Figure 3) is illustrated as a mechanism-based inhibitor of CYP2C19, but also as a strong substrate of CYP3A4.

For clarity, the legends on Figures 3, 7, 8, 9, and 10 have been enlarged to make sure that the reader understands and interpret easily the color code of individual boxes. In addition, we have modified the text in the legend as follows; “weak substrates, moderate substrates, and strong substrates along with inhibitors and mechanism-based inhibitors.”

  • For the example in Pages 4-5, it will be good to mention it is an IV drug, since gut DDI is not considered.

This is an interesting comment as, conceptually, both the renal and metabolic clearances are calculated as Amount Excreted [Ae of either the parent drug (renal clearance) or of the metabolites (for metabolic clearance)] over the AUC of the parent compound. So, the sum of the renal clearance and metabolic clearance corresponds to CL/F. Obviously, when the drug is administered IV, F=1 and the estimated clearance is the systemic clearance. For drugs with a high bioavailability, this equation is independent of the route of administration. But for drugs extensively metabolized in the intestine during first pass (low Fg), the relative contribution of the renal and metabolic clearances in the disposition of a drug will change according to the routes of administration. In the example presented, a relative contribution of the renal and metabolic clearance to the total clearance of the drug is indicated (values are presented as a fraction of total clearance equal to 1). Under these conditions, the equation shown is appropriate and is “route of administration” independent since it presents relative changes in total clearance.

  • Equation (9), it should be CL = Dose / AUC

For clarity, we have added the more information to equation 9, as follows:

CL = Dose / AUC0-∞

Under steady-state conditions, the area under the drug concentration curve (AUC) measured over a dosing interval (t) is equal to AUC0-∞.  Since the average concentration over a dosing interval (Cav) at steady-state can be estimated by the AUC0-t / t, the equation could be rearranged in a simpler manner to yield:

CL = Dose / (Cav * t)

Reviewer 2 Report

The manuscript is well written and comprehensive. However, in genomic era, it is necessary to dedicate a short chapter to very important aspect of genetic polymorphisms of CYP450 enzymes and role of CYP450 pharmacogenetics in clinical practice.

Author Response

We thank the reviewer for taking the time to review and provide positive and productive comments. We have addressed these comments and made appropriate changes:

  • The manuscript is well written and comprehensive. However, in genomic era, it is necessary to dedicate a short chapter to very important aspect of genetic polymorphisms of CYP450 enzymes and role of CYP450 pharmacogenetics in clinical practice.

We agree with this comment from Reviewer 2 pointing out the contribution of CYP450 genetic polymorphisms in intersubject variability in drug response. As suggested by the reviewer, a short paragraph was added to highlight the importance of CYP450 genetic polymorphisms and their influence in drug-induced phenoconversion and on the magnitude of drug-drug-gene interactions (lines 451-462). 

“In addition to drug-drug interactions, high variability in terms of CYP450 expression and/or activities can be explained by genetic polymorphisms in genes encoding specific isoforms (such as CYP2C9, CYP2C19 and CYP2D6). This variability on CYP450 expression/activities translates into intersubject variability in drug disposition and drug response. Often, the impact of genetic polymorphisms and drug-drug interactions on CYP450s have been studied separately. However, an interaction exists between these factors. Genetic polymorphisms could also contribute to variability observed in the magnitude of drug-drug interactions observed between two drugs. So, genetic polymorphisms in drug metabolizing enzymes can affect the occurrence of phenoconversion induced by drug inhibitors. As reported by Storelli et al. differences in CYP2D6 inhibition observed in vitro with paroxetine (MBI) or duloxetine (competitive inhibitor) across CYP2D6 genotypes were not related to their inhibition parameters but likely due to a differential level of functional enzymes as a function of CYP2D6 genotype.”1, 2

Reviewer 3 Report

Authors have presented a review of mechanism based inhibition of cytochromes P450 enzymes in relation to drug-drug interactions. Topic is interesting, although many authoritative reviews exist, written by some of the leading expert in the field.

Manuscript is well written and concepts are explained with graphics to depict the different modes of inhibition of CYP enzymes.

Minor points:

  1. Lines 58-62: Define the reaction of P450 enzymes in proper context. P450 chemical reactions do not use NADPH directly, either mitochondrial Adrenodoxin-Adrenodoxin reductase system (See reviews by Rita Bernhardt and colleagues, and by T Omura) or P450 reductase system (see review by Amit V Pandey and colleagues) serves as a redox partner to P450s for the supply of electrons from NADPH.
  2. Lines 106 onward: In reality, it is mixed inhibition pattern that is often seen in enzyme assays and in vivo studies. Some discussion of this needs to be included earlier (couple of lines about material mentioned in lines 207-217).
  3. Lines 188-201: While non competitive inhibition in other enzymes can be purely from binding to allosteric sites, in P450 systems, interruption of P450-CYB5, and P450-POR (NADPH Cytochrome P450 reductase) does have a role. NADPH is often still used, it is only the transfer of electrons to P450s is interrupted, and NADPH is wasted in this instance (coupling-uncoupling of P450-POR).
  4.  In the conclusions expansion of the practical examples covered in detail can be summarized.

Author Response

We thank the reviewer for taking the time to review and provide positive and productive comments. We have addressed these comments and made appropriate changes:

  • Authors have presented a review of mechanism-based inhibition of cytochromes P450 enzymes in relation to drug-drug interactions. Topic is interesting, although many authoritative reviews exist, written by some of the leading expert in the field.

Although other publications exist on this topic, they are often presented at a high level for experts in the area and lack a needed didactic nature. In preparing this manuscript, we have taken good care to present scientifically valid information and support the concepts with examples.

  • Manuscript is well written and concepts are explained with graphics to depict the different modes of inhibition of CYP enzymes.

We thank the reviewer for this comment.

  • Minor points:
  1. Lines 58-62: Define the reaction of P450 enzymes in proper context. P450 chemical reactions do not use NADPH directly, either mitochondrial Adrenodoxin-Adrenodoxin reductase system (See reviews by Rita Bernhardt and colleagues, and by T Omura) or P450 reductase system (see review by Amit V Pandey and colleagues) serves as a redox partner to P450s for the supply of electrons from NADPH.

We agree with this reviewer’s comment about the complex nature of redox partners used by the CYP450 system. To keep the narrative simple and focused, the sentence has been slightly modified as follows (lines 62-63):

For CYP450 isoforms, binding to the active site is independent of the NADPH-P450 oxidase reactions; however, the chemical reaction leading to the formation of the metabolite will employ electrons originated from NADPH.

  1. Lines 106 onward: In reality, it is mixed inhibition pattern that is often seen in enzyme assays and in vivo studies. Some discussion of this needs to be included earlier (couple of lines about material mentioned in lines 207-217).

As suggested by the reviewer, the text has been changed as follows (lines 98-99):

Drugs defined as inhibitors bind either to the active site or to an allosteric site of the enzyme. However, they can also bind to both; in these cases, the process is called ‘mixed inhibition’ and can often be more potent than simple competitive or non-competitive inhibition.

  1. Lines 188-201: While non-competitive inhibition in other enzymes can be purely from binding to allosteric sites, in P450 systems, interruption of P450-CYB5, and P450-POR (NADPH Cytochrome P450 reductase) does have a role. NADPH is often still used, it is only the transfer of electrons to P450s is interrupted, and NADPH is wasted in this instance (coupling-uncoupling of P450-POR).

We agree with this reviewer that cofactor systems like CYPb5 and P450-POR play a very important role in regulating CYP450 isoform activities. This has been extensively shown in in vitro and animal studies. A sentence has been added at the end of section 3.1.2 (lines 211-213):

Other non-competitive inhibition conditions may involve CYPb5 and/or CYP450 oxidoreductase as these factors have been shown to modulate CYP450 activities, at least in in vitro systems.3, 4

  1. In the conclusions expansion of the practical examples covered in detail can be summarized.

As requested by the reviewer, we have made several changes in the conclusion section to summarize the relevance of drug-drug interactions due to MBI by highlighting some clinical examples discussed in the review, especially in the context of polypharmacy.

Polypharmacy in many cases is deemed required and elderly patients are particularly prone to this phenomenon. Aging is associated with the presence of multiple independent chronic diseases and is almost always accompanied by multiple drug regimens. Polypharmacy has been associated with many adverse clinical outcomes such drug-drug interactions leading to adverse drug events. Among these, mechanism-based inhibitor-victim drug combinations like clopidogrel and omeprazole are commonly prescribed. Drugs for overactive bladder like mirabegron and tricyclic antidepressants like paroxetine are also commonly prescribed in elderly populations. Polypharmacy is not necessarily synonymous with inappropriate treatment, but in several situations, it can lead to significant drug-drug interactions especially in presence of mechanism-based inhibitor drugs as described in the current review. In these cases, polypharmacy might cause problems like blunted efficacy of clopidogrel due to the co-administration of omeprazole, or increased toxicity of other drugs co-administered with paroxetine or mirabegron. Clinicians must be able to recognize and intervene appropriately based on the mechanism of these interactions.

  1. Storelli F, Desmeules J, Daali Y. Genotype-sensitive reversible and time-dependent CYP2D6 inhibition in human liver microsomes. Basic & Clinical Pharmacology & Toxicology. 2019;124(2):170-80.
  2. Storelli F, Matthey A, Lenglet S, Thomas A, Desmeules J, Daali Y. Impact of CYP2D6 Functional Allelic Variations on Phenoconversion and Drug-Drug Interactions. Clin Pharmacol Ther. 2018;104(1):148-57.
  3. Zhang H, Gao N, Liu T, Fang Y, Qi B, Wen Q, et al. Effect of Cytochrome b5 Content on the Activity of Polymorphic CYP1A2, 2B6, and 2E1 in Human Liver Microsomes. PloS one. 2015;10(6):e0128547-e.
  4. Bart AG, Scott EE. Structural and functional effects of cytochrome b(5) interactions with human cytochrome P450 enzymes. J Biol Chem. 2017;292(51):20818-33.